# Effect of School-Based Nutrition and Health Education for Rural Chinese Children

**DOI:** 10.3390/nu14193997

**Published:** 2022-09-27

**Authors:** Ying Xu, Xiaoyi Bi, Tingting Gao, Titi Yang, Peipei Xu, Qian Gan, Juan Xu, Wei Cao, Hongliang Wang, Hui Pan, Zhibin Ren, Chunjie Yin, Qian Zhang

**Affiliations:** 1Chinese Center for Disease Control and Prevention, NHC Key Laboratory of Trace Element Nutrition, National Institute for Nutrition and Health, Beijing 100050, China; 2Beijing Tongzhou District Center for Disease Control and Prevention, Beijing 101199, China; 3Beijing Shunyi District Center for Disease Control and Prevention, Beijing 101300, China; 4School of Public Health, Xinjiang Medical University, Urumqi 830017, China

**Keywords:** children, nutrition knowledge, dietary behaviors, nutrition education, school-based intervention

## Abstract

The nutritional status of rural Chinese children has improved in recent years, but their nutritional knowledge is still relatively lacking. School-based nutrition and health education was conducted for children in three counties of China from 2018 to 2020. The students in the intervention schools were given two-year nutrition and health education courses, while the control schools did not receive any intervention. Students’ nutrition knowledge, dietary intake, and dietary behaviors were collected using a questionnaire, and height and weight were measured uniformly. The nutrition knowledge score in the intervention group was increased by 1.01 and 0.64 points in the first and second years. A multilevel model was used to evaluate the intervention effects. Statistically significant interactions between groups and time were observed in nutrition knowledge, the frequency of eating breakfast, and dietary intake, including meat, eggs, milk, and vegetables (*p* < 0.05), but not in nutritional status. Therefore, the supplementation of school-based nutrition and health education had a positive impact on the nutrition knowledge and dietary intake of rural Chinese children.

## 1. Introduction

Childhood and adolescence are key periods of growth and development and are also critical periods for cultivating healthy dietary behaviors. With economic development in recent years, the nutritional status of rural Chinese children has continuously improved [1]. However, the nutritional knowledge of children is still relatively insufficient, which affects their physical and intellectual development [2,3]. There is evidence from studies in some countries that school-based nutrition and health education can effectively improve nutrition knowledge and develop healthy dietary behaviors in children [4,5,6,7]. For instance, Vered et al. conducted a school-based nutrition knowledge intervention for six months in children aged 4–7 years in Israeli schools of low socioeconomic status, which significantly improved nutrition knowledge and increased the food variety of children in the intervention group [8].

Chinese children have experienced a dramatic shift from traditional dietary patterns (mainly grains, vegetables, and tubers) to Western dietary patterns (mainly desserts, fast food, and meat) [9]. The consumption of beverages has become increasingly common among Chinese children aged 6–17 years [10]. Therefore, it is imperative to strengthen the nutrition and health education of rural children. The China Youth Development Foundation promoted and implemented the Hope Kitchen Plan in rural schools in Guangxi and Hubei Province from 2018 to 2020. This study aimed to evaluate the effects of school-based nutrition and health education on children’s nutrition knowledge, dietary behaviors, dietary intake, and nutritional status.

## 2. Materials and Methods

### 2.1. Participants

Three counties were selected for this study: Zigui County in Hubei Province and Du‘an County and Long‘an County in Guangxi Province, China. The GDP of these three counties was CHY 36,682.85 (USD 5451.07), CHY 16,468.00 (USD 2447.14), and CHY 24,169.00 (USD 3591.51) per person in 2018, respectively, while the national average GDP was CHY 65,534.00 (USD 9738.35) [11,12].

A total of 15 rural primary schools were selected, including four schools in Long’an county, five schools in Zigui County, and six schools in Du’an County. Nine schools were selected as the intervention group. Six schools were recruited as the control group. These schools were similar in size, school facilities, and student composition to the schools in the intervention group. There were two control schools in each county and two intervention schools in Long’an county, three intervention schools in Zigui County, and four intervention schools in Du’an County.

All children from grades 2 to 4 (8–10 years) were recruited at baseline. There were 2655 children (grades 2–4), 2567 children (grades 3–5), and 2503 children (grades 4–6) in September 2018, September 2019, and December 2020, respectively. The reason for being unable to follow up (5.7%) was the graduation of children. A total of 2066 children who participated in all three surveys were selected. Children had the right to refuse to participate in the study, and no one refused to participate (Figure 1).

### 2.2. Intervention Methods

Children in the intervention schools were provided nutrition education with nutrition and health courses as the main measure for two consecutive academic years from September 2018, after the baseline survey, to June 2020, before the final survey. Nutrition and health education courses were taught to children by professionally trained teachers as daily courses. We also carried out nutrition-related activities on campus. The main interventions included the following. 

A *nutrition class* series of textbooks (including two student books and one teacher book) and electronic courseware were used to provide nutrition and health courses for two consecutive academic years in the intervention schools, with one 40 min class every two weeks and five to six classes per semester. The main contents of the textbooks included food, nutrients, and dietary behaviors.Organizing unified training for teachers of nutrition courses, including four face-to-face training sessions and three online training sessions in total. The main contents included basic knowledge of nutrition, interpretation of the Chinese Dietary Guidelines, nutrition deficiency and dietary prevention for rural children, and the national nutrition policy for children.Holding the *nutrition class* competition for teachers, the *nutrition class* essay, painting and speech competitions for children, the Healthy Life Weekly Notes during winter and summer holidays, and other nutrition promotion and education activities.Providing physical activity resources that can improve the convenience and enthusiasm of children participating in sports such as basketball and skipping rope.Organizing children plating vegetables themselves, which not only improved children’s awareness of increasing intake of fresh fruits and vegetables but also became a labor practice base for students to understand nature.

Children in the control schools received their usual curriculum and did not receive any intervention in the nutrition education or physical activities.

### 2.3. Data Collection 

Data related to the date of birth, sex, grade, area, nutritional knowledge, dietary intake, dietary behaviors, and physical activities were collected using a student questionnaire based on the China National Nutrition and Health Surveillance [13]. Children completed the questionnaires by themselves after the investigators explained them to the children in detail. Data were collected at baseline, after the first and second years. 

Nutrition knowledge: The questionnaire consisted of 10 questions. Each correct response was assigned one point, and an incorrect or no answer was assigned 0 points. The total knowledge score ranged from 0 to 10 points with a higher score indicating a higher level of nutrition knowledge. 

Dietary intake: The questionnaire consisted of five questions, including milk and egg consumption, the frequency of consumption of meat and fruits, and the variety of vegetables consumed in the past week. The total dietary intake score ranged from 0 to 15 points. A higher score indicates a healthier dietary intake (see Table 1).

Dietary behaviors: The questionnaire consisted of four questions on the frequency of eating breakfast, snacks, beverages, and plain water in the past week. The total score for dietary behaviors ranged from 0 to 12 points. Higher scores indicate healthier dietary behaviors.

The children’s fasting height and weight were examined early in the morning. Weight was measured to the nearest 0.1 kg in light indoor clothing, and height was measured without shoes to the nearest 0.1 cm. Body mass index (BMI) (kg/m^2^) was calculated by dividing weight (kg) by height squared (m^2^). Nutritional status was based on BMI by age and sex and divided into stunting, wasting, normal, bodyweight, and obesity. Malnutrition, including stunting and wasting, was screened according to the Chinese Screening Standard for Malnutrition in School Children and Adolescents (WS/T456-2014) [14]. Overweight and obesity screening was conducted according to the Chinese Screening for Overweight and Obesity among school-age children and adolescents (WS/T586-2018) [15]. 

This study was approved by the Ethics Committee of the China Center for Disease Control and Prevention. All participants provided informed consent prior to participating in the study.

### 2.4. Statistical Analyses

All statistical analyses were performed using SAS (SAS 9.4 for Windows, SAS Institute, Inc., Cary, NC, USA). Means ± SDs were used to describe quantitative data, and qualitative data were summarized as percentages.

This study measured each subject for three consecutive years; therefore, the three measurements of the same subject were not independent. A multilevel model was used to evaluate the effect of the intervention, and time was used as a level 1 variable to explain the difference in outcome indicators of the control group at baseline in the first and second year. Taking the individual as the level 2 variable, we included the group in the model as a fixed effect to explain the difference in outcome indicators between the intervention and control groups at baseline. The interaction effect between time and group explains the effects of the intervention. All *p*-values < 0.05 were considered to indicate statistical significance.

## 3. Results

### 3.1. The Characteristics of the Participants

A total of 2066 children were enrolled in this study, which included 1077 boys and 989 girls at baseline. There were 1563 children in the intervention group and 503 children in the control group. The average age of children was 9.0 years at baseline. The number of children in grades 2, 3, and 4 was 682 (33.0%), 662 (32.0%), and 722 (35.0%), respectively. There were 837 (40.5%), 875 (42.4%), and 354 (17.1%) children in Zigui County, Du Unk County, and Long Unk County, respectively. The number of children whose physical activity times were 0–30, 30–60, and ≥60 min/day was 603 (29.2%), 722 (35.0%), and 741 (35.9%), respectively (see Table 2).

### 3.2. Comparison of Nutrition Knowledge, Dietary Intake, and Dietary Behavior Scores at Baseline, First, and Second Year

#### 3.2.1. Nutrition Knowledge

The correct rate of nutrition knowledge at baseline was 10.0–73.5% in the intervention group, compared with 6.0–69.8% in the control group. After the two-year nutrition and health courses, the correct rate in first and second year was 14.6–84.5% and 21.2–93.0% in the intervention group, respectively. In the second year, except for the correct rate of the nutritional characteristics of coarse grains (−13.0%), the correct rate of other nutrition knowledge in the intervention group increased compared with the baseline, with an increasing range of 5.1–39.8% (see Table 3).

The nutrition knowledge score in the intervention group was increased by 1.01 and 0.64 points in the first and second years, compared with the control group (0.70 and 0.57 points), respectively (Figure 2). The results of the multilevel model showed that the interaction effect between time and group was statistically significant in the first year (*p* < 0.05) and marginally significant in the second year (*p* = 0.068) (Table 4).

#### 3.2.2. Dietary Intake

Table 5 presents the differences (intervention versus control) from baseline to the second year in milk and egg consumption, frequency of meat and fruit consumption, and variety of vegetables among children. The multilevel model showed that the interaction effect between time and group of milk, meat, eggs, and vegetables was statistically significant in the second year (*p* < 0.05). The frequency of fruit consumption in the intervention group increased compared to the baseline, but there was no statistical significance in the interaction effect between time and group of fruits (*p* > 0.05).

The dietary intake score in the intervention group was increased by 0.33 and 0.13 points in the first and second years compared with the control group (−0.04 and 0.05 points), respectively (Figure 3). The results of the multilevel model showed that the interaction effect between time and dietary intake group was statistically significant in the first and second year (*p* < 0.05) (Table 6).

#### 3.2.3. Dietary Behaviors

Table 7 presents the differences (intervention versus control) from baseline to the second year in the frequency of breakfast, snack, beverage, and plain water consumption among children. The multilevel model showed that the interaction effect between time and breakfast group was statistically significant in the first year but not in the second year (*p* < 0.05). There was no statistically significant interaction effect between time and group of snacks, beverages, or plain water (*p* > 0.05) (Table 7).

The dietary behavior score in the intervention group was increased by 0.19 and 0.15 points in the first and second years compared with the control group (0.11 and 0.13 points, respectively) (Figure 4). There was no statistically significant interaction effect between time and group of dietary behaviors (*p* > 0.05) (Table 8). 

### 3.3. Comparison of Height, Weight, BMI, and Nutritional Status at Baseline, First, and Second Year

The proportion of stunting was 6.1% in the intervention group and 10.3% in the control group at baseline and 3.1% and 7.6% in the second year, respectively. The proportion of obesity was 8.0% in the intervention group and 3.4% in the control group at baseline and 7.7% and 3.8% in the second year, respectively (Figure 5 and Figure 6). The results of the multilevel model showed that there was no significant difference in the interaction effect between time and height, weight, BMI, and nutritional status between the intervention and control groups (*p* > 0.05) (Table 9 and Figure 5 and Figure 6).

## 4. Discussion

The present study evaluated the effectiveness of school-based nutrition and health education on nutrition knowledge, dietary intake, dietary behaviors, and nutritional status among rural Chinese children. The results of this study suggest that school-based nutrition and health education may have a positive effect on nutrition knowledge, the frequency of eating breakfast, and dietary intake, including meat, eggs, milk, and vegetables, but not on nutritional status.

Countries worldwide have attached importance to nutrition and health education for children [16,17]. The World Health Organization proposed the Nutrition Friendly Schools Initiative in 2006, advocating comprehensive measures of school-based nutrition and health education [18]. In recent years, a wide range of nutrition education interventions have also been carried out for children in countries such as the United States, China, and France, which effectively improved children’s nutrition knowledge [7,8,9,10,19]. Marwa et al. conducted a 6-month school-based nutrition intervention on Syrian refugee children aged 6–14 years in Bekaa, Lebanon. The intervention included educational courses and the provision of local healthy snacks. They found that dietary knowledge in the intervention group (*β* = 1.22, 95% CI: 0.54–1.89) increased significantly compared to the control group (*p* < 0.05) [4]. Our research also observed that nutrition knowledge increased by 1.01 points in the intervention group in the first year compared with 0.70 points in the control group (*p* < 0.05), suggesting that school-based nutrition and health education may contribute to the comprehension of children’s nutrition knowledge. However, we found that awareness of the nutritional characteristics of coarse grains decreased. The possible cause was the low intake of coarse grains in rural Chinese children, making them pay less attention to relevant nutrition knowledge. The intake of coarse grains in rural children aged 6–11 years was only 12.1 g/d in the Report on Nutrition and Chronic Diseases of Chinese Residents (2020) [20], which is much lower than the recommended intake of 30–70 g/day for this age group in the Chinese Dietary Guidelines for School-Aged Children [21]. Our results revealed that the teaching contents and methods should be adjusted according to the dietary characteristics of subjects; therefore, nutrition education on coarse grains needs to be further strengthened in the future.

A reasonable dietary structure is critical to ensure children’s nutrition and health. Our study observed that only 44.4%, 14.3%, and 28.3% of children in the intervention group consumed meat, eggs, and milk every day at baseline, which was far below the recommended intake and frequencies [22]. After two years of nutrition and health education, children’s milk and egg consumption, the frequency of meat consumption, and the variety of vegetables consumed in the intervention group improved and were significantly higher than those the control group in the second year. Our results are consistent with those of other studies in the United States, Asia, and Iran [22,23,24]. This indicates that nutrition and health education can contribute to a rational diet for children.

Breakfast can provide the body with essential nutrients and energy, which is important for the health of children [25]. Our study found that nutritional knowledge related to breakfast and the frequency of eating breakfast improved after the intervention. This showed that the improvement of children’s nutrition knowledge can improve their dietary behaviors to a certain extent. Nevertheless, 21.3% of the intervention group and 33.2% of the control group still failed to eat breakfast every day of second year. Other studies have revealed that skipping or eating breakfast irregularly may not only lead to malnutrition in children but also increase the risk of obesity and other related chronic diseases [26,27,28]. Therefore, education on the importance of breakfast among rural children should be further enhanced.

Similar to the results observed in South Africa and Asia [29,30], our results showed that the behavior of consuming snacks and beverages did not significantly improve after the intervention. This may be related to the current widespread consumption of snacks and beverages among Chinese children [31]. Reports on the consumption of sugar-sweetened beverages by Chinese children have pointed out that the production and consumption of beverages in China have increased rapidly in recent years [32]. In addition, poor self-control in children is also a major cause of snack and beverage consumption [33]. It may also be influenced by family and societal factors such as personal preferences, advertising, and marketing [34].

There were no significant differences in BMI and nutritional status between the intervention and control groups in this study. The United States, Australia, and many European countries have widely adopted school-based interventions to reduce weight by improving children’s nutrition knowledge and changing lifestyles, but their effectiveness is different [35,36]. A recent systematic review showed that school-based prevention interventions are mildly effective in reducing BMI in children. These latest studies tend to be more comprehensive and longer and include more factors, such as environmental modification, diet improvement, and parental support [37]. The different results may be related to the lack of intervention for other factors and confounders in our study. The detailed reasons for this difference require further discussion.

This study has several strengths and limitations. The strengths of this study include having a relatively large sample size, a design with a comparison group, and high adherence rates. One limitation was that the intervention only lasted for two years and did not observe the long-term influence on children’s dietary behaviors and nutritional status.

## 5. Conclusions

Our findings suggest a promising impact of integrated nutrition health education on nutrition knowledge, the frequency of eating breakfast, and dietary intake of meat, eggs, milk, and vegetables of rural Chinese children.

It is still necessary to explore scientific and long-term nutrition knowledge and behavioral intervention models for children and adolescents. Future studies are needed to test the feasibility of scaling up such nutritional interventions and also to evaluate their long-term impact on children’s dietary behaviors and nutritional status.

## Figures and Tables

**Figure 1 nutrients-14-03997-f001:**
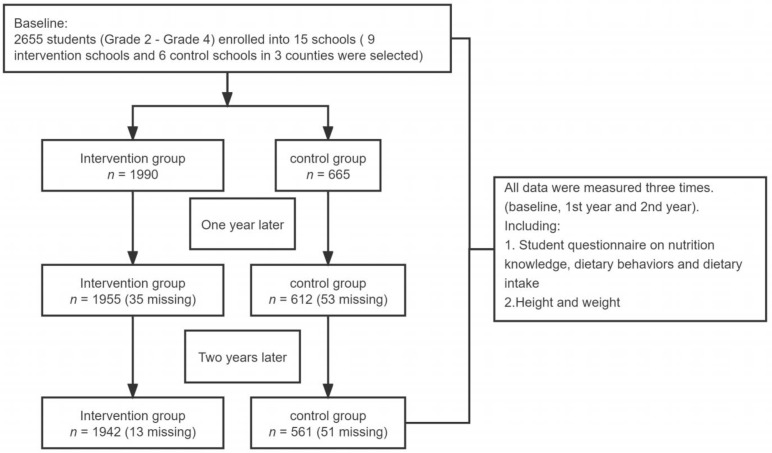
Flow diagram of participation.

**Figure 2 nutrients-14-03997-f002:**
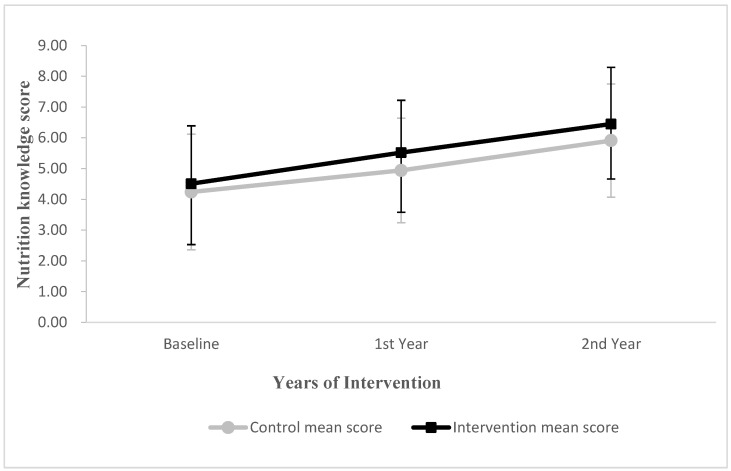
Changes in mean score for children’s nutrition knowledge in first and second year from baseline. Vertical bars indicate standard deviations.

**Figure 3 nutrients-14-03997-f003:**
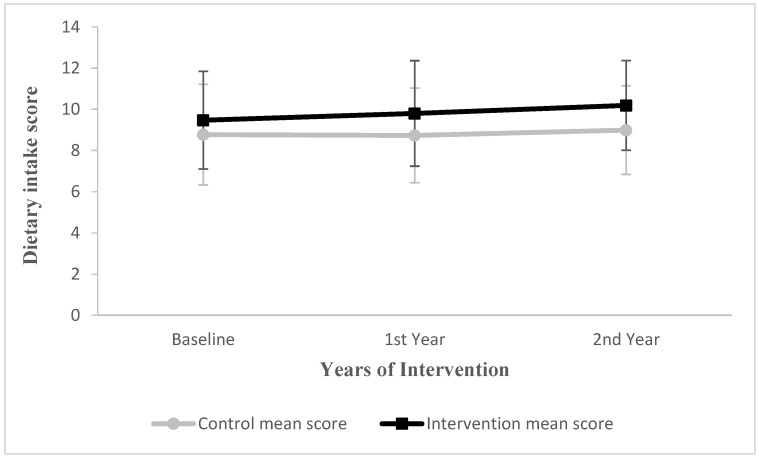
Changes in mean score for children’s dietary intake in first and second year from baseline. Vertical bars indicate standard deviations.

**Figure 4 nutrients-14-03997-f004:**
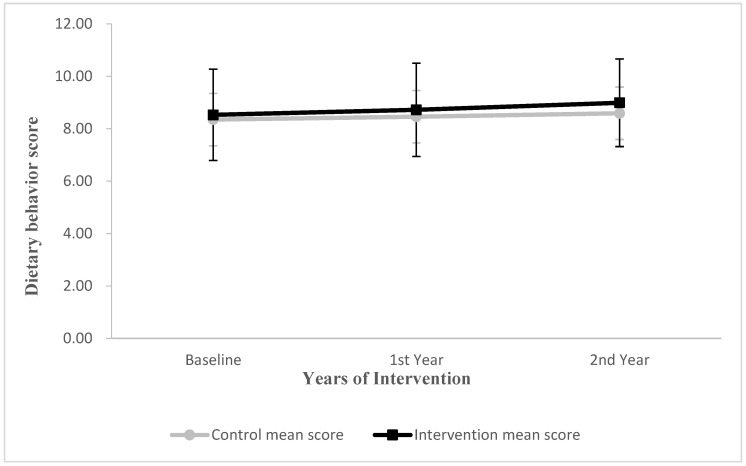
Changes in mean score for children’s dietary behaviors in first and second year from baseline. Vertical bars indicate standard deviations.

**Figure 5 nutrients-14-03997-f005:**
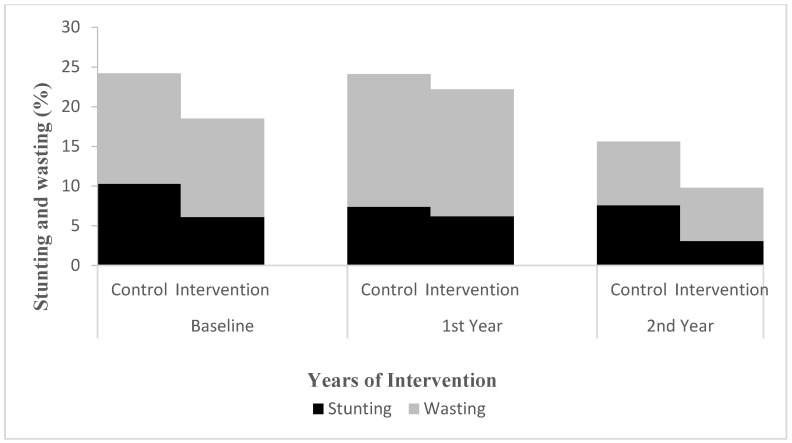
Changes in proportion for children’s stunting and wasting in first and second year from baseline. Multilevel model was used to evaluate the effect of the intervention after adjusting for physical activity. Stunting: group effect, *p* < 0.05. Wasting: time effect (two years), *p* < 0.05. Others all *p* > 0.05.

**Figure 6 nutrients-14-03997-f006:**
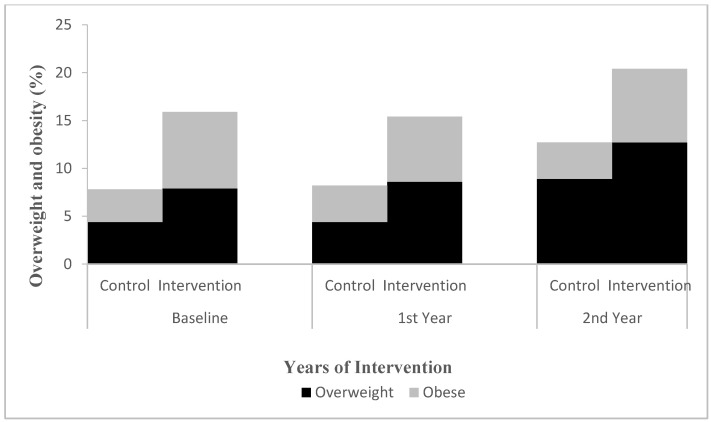
Changes in proportion for children’s overweight and obesity in first and second year from baseline. Multilevel model was used to evaluate the effect of the intervention after adjusting for physical activity. Overweight: group effect: *p* < 0.05; time effect (two years): *p* < 0.05. Obesity: group effect, *p* < 0.05. Others all *p* > 0.05.

**Table 1 nutrients-14-03997-t001:** Points assigned to each variable of dietary intake and dietary behaviors.

	Variable	Points
Dietary intake		
1	Milk consumption	Less than 1 bag/week = 0; 1–3 bags/week = 1 point; 4–6 bags/week = 2 points; 1 bag/day and above = 3 points
2	Meat consumption frequency	Less than 1 time/week = 0; 1–3 times/week = 1 point; 4–6 times/week = 2 points; 1 time/day and above = 3 points
3	Egg consumption	Less than 1/week = 0; 1–3/week = 1 point; 4–6/week = 2 points; 1/day and above = 3 points
4	Vegetable consumption variety	Less than 1 kind/week = 0; 1 kind/day = 1 point; 2 kinds/day = 2 points; 3 kinds/day and above = 3 points
5	Fruit consumption frequency	Less than 1 time/week = 0; 1–3 times/week = 1 point; 4–6 times/week = 2 points; 1 time/day and above = 3 points
Dietary behaviors		
6	Breakfast consumption frequency	1–2 days/week = 0; 3–4 days/week = 1 point; 5–6 days/week = 2 points; everyday = 3 points
7	Snack consumption frequency	Less than 1 time/week = 3 points; 1–3 times/week = 2 points; 4–6 times/week = 1 point; 1 time/day and above = 0
8	Beverage consumption frequency	Less than 1 time/week = 3 points; 1–3 times/week = 2 points; 4–6 times/week = 1 point; 1 time/day and above = 0
9	Plain water consumption	Less than 1 cup/day = 0; 1–2 cups/day = 1 point; 3–4 cups/day = 2 points; 5 cups/day and above = 3 points

Note: The volume of milk is 250 mL. The weight of an egg is 50~60 g. The volume of a cup of water is about 300 mL.

**Table 2 nutrients-14-03997-t002:** Comparison of characteristics of children from the intervention group and control group at baseline.

	Total Sample (*n* = 2066)	Control (*n* = 503)	Intervention (*n* = 1563)
Age (years), mean ± SD	9.0 ± 1.04	9.2 ± 1.11	8.9 ± 1.01
Gender, *n* (%)			
Male	1077 (52.1)	268 (53.3)	809 (48.2)
Female	989 (47.9)	235 (46.7)	754 (51.8)
Grade, *n* (%)			
Two	682 (33.0)	180 (35.8)	502 (32.1)
Three	662 (32.0)	139 (27.6)	523 (33.5)
Four	722 (35.0)	184 (36.6)	538 (34.4)
County, *n* (%)			
Zigui County	837 (40.5)	172 (34.2)	665 (42.6)
Du‘an County	875 (42.4)	240 (47.7)	635 (40.6)
Long‘an county	354 (17.1)	91 (18.1)	263 (16.8)
Physical activity, *n* (%)			
0–30 min/d	603 (29.2)	196 (39.0)	407 (26.0)
30–60 min/d	722 (35.0)	146 (29.0)	576 (36.9)
≥60 min/d	741 (35.9)	161 (32.0)	580 (37.1)

**Table 3 nutrients-14-03997-t003:** Comparison of correct rate of children’s nutrition knowledge between the intervention group and control group (%).

	Baseline		First Year		Second Year		Change (Baseline-Second Year)	
	Control	Intervention	Control	Intervention	Control	Intervention	Control	Intervention
Health is not only the absence of disease but also good psychological and social adaptability ^#,@,@@^	30.8	45.0	58.3	67.1	73.2	84.8	42.4	39.8
The most abundant protein is meat, poultry, fish and eggs ^@,@@^	26.8	30.5	42.3	47.7	62.2	63.6	35.4	33.1
The best source of calcium is milk ^@@,^*	37.6	40.4	41.7	52.3	58.4	62.1	20.8	21.8
Iron deficiency anemia can be prevented by eating more lean meat and vegetables^@,@@^	48.7	49.3	62.8	63.1	70.0	75.5	21.3	26.2
A nutritious breakfast should include four types of food ^#,@@,^**	6.0	10.0	6.8	14.6	10.3	25.7	4.3	15.7
China recommends that school-age children drink more than 300 g of milk and dairy products every day ^#,@,^*^,^**	21.5	16.1	10.9	17.0	16.7	21.2	−4.8	5.1
Fresh vegetables and fruits cannot be substituted for each other ^@,@@,^**	51.7	54.3	65.0	68.1	68.8	79.8	17.1	25.5
Coarse grains have more comprehensive nutritional characteristics than fine grains ^@,^**	66.4	65.4	53.9	56.9	63.0	52.4	−3.4	−13.0
Obese children are more prone to hypertension, hyperlipidemia, and other diseases: yes ^@@,^*^,^**	69.8	66.6	73.4	80.4	83.3	87.5	13.5	20.9
Food not less likely to deteriorate when put in the refrigerator ^#,@,@@^	64.4	73.5	78.7	84.5	85.5	93.0	21.1	19.5

Note: A multilevel model was used to evaluate the effect of the intervention. Group effect: ^#^ *p* < 0.05; time effect: ^@^ one year *p* < 0.05; ^@@^ two years *p* < 0.05; time × group effects: * *p* < 0.05; ** two years *p* < 0.05.

**Table 4 nutrients-14-03997-t004:** Test of the fixed effects of various factors of nutrition knowledge score.

Effect	Type	*β* *	SE ^#^	T-Value	*p*-Value
Intercept			0.083	51.16	<0.001
Group	Intervention		0.096	2.10	0.007
	Control	Ref.			
Time	Second Year		0.111	15.40	<0.001
	First Year		0.113	5.97	<0.001
	Baseline	Ref.			
Time × Group	Second Year × Intervention		0.128	1.83	0.068
	Second Year × Control				
	First Year × Intervention		0.129	2.44	0.015
	First Year × Control				
	Baseline × Intervention	Ref.			
	Baseline × Control				

Note: * *β*, coefficient. ^#^ SE, standard error.

**Table 5 nutrients-14-03997-t005:** Comparison of milk consumption; the frequency of meat, egg, and fruit consumption; and the variety of vegetables of children in the past week between the intervention group and control group (%).

Consumption/Frequency/Variety	Baseline		First Year		Second Year	
	Control	Intervention	Control	Intervention	Control	Intervention
Milk ^#,@,^**						
Less than 1 bag/week	14.3	7.9	7.6	11.4	6.0	3.8
1–3 bags/week	45.5	51.2	47.7	38.8	53.5	39.2
4–6 bags/week	19.5	12.5	22.7	13.5	26.6	18.4
1 bag/day and above	20.7	28.3	22.1	36.3	13.9	38.6
Meat ^@,@@,^*^,^**						
Less than 1 time/week	1.8	2.7	3.6	2.0	1.2	1.5
1–3 times/week	39.0	36.6	45.5	40.2	37.6	30.5
4–6 times/week	12.7	16.3	19.7	18.1	24.7	23.8
1 time/day and above	46.5	44.4	31.2	39.7	36.6	44.1
Eggs *^,^**						
Less than 1/week	11.9	6.5	10.5	6.5	8.3	4.4
1–3/week	54.3	61.9	62.4	52.0	62.8	49.6
4–6/week	20.1	17.2	17.3	21.2	18.5	29.4
1/day and above	13.7	14.3	10.7	20.4	10.3	16.6
Vegetables ^#,@,@@,^**						
Less than 1 kind/week	1.4	0.3	1.2	0.8	0.2	0.4
1 kind/day	18.3	9.0	13.5	6.5	9.5	3.9
2 kinds/day	27.6	20.7	24.9	19.3	25.8	21.8
3 kinds/day and above	52.7	70.1	60.4	73.4	64.4	73.9
Fruits ^#^						
Less than 1 time/week	3.0	1.7	3.0	3.0	3.0	2.2
1–3 times/week	55.5	42.6	49.5	36.6	47.5	37.5
4–6 times/week	20.5	25.9	29.4	28.7	27.6	28.6
1 time/day and above	21.1	29.8	18.1	31.7	21.9	31.7

Note: Multi-level model was used to evaluate the effect of the intervention after adjusting the nutrition knowledge score. Group effect: ^#^ *p* < 0.05; time effect: ^@^ one year *p* < 0.05; ^@@^ two years *p* < 0.05; time × group effects: * *p* < 0.05; ** two years *p* < 0.05.

**Table 6 nutrients-14-03997-t006:** Test of the fixed effects of various factors of dietary intake score.

Effect	Type	*β* *	SE ^#^	T-Value	*p*-Value
Intercept			0.104	83.94	<0.001
Group	Intervention		0.120	5.95	<0.001
	Control	Ref.			
Time	Second Year		0.138	1.19	0.235
	First Year		0.140	−0.77	0.441
	Baseline	Ref.			
Time × Group	Second Year × Intervention		0.159	3.45	<0.001
	Second Year × Control				
	First Year × Intervention		0.160	2.68	0.007
	First Year × Control				
	Baseline × Intervention	Ref.			
	Baseline × Control				

Note: After adjusting for the nutrition knowledge score, group effect *p* < 0.05; time effect *p* > 0.05; time × group effect *p* < 0.05. * *β*, coefficient. ^#^ SE, standard error.

**Table 7 nutrients-14-03997-t007:** Comparison of the frequency of breakfast, snack, beverage, and plain water consumption of children in the past week between the intervention group and control group (%).

Consumption/ Frequency	Baseline		First Year		Second Year	
	Control	Intervention	Control	Intervention	Control	Intervention
Breakfast ^#,@,^*						
1–2 days/week	9.5	6.1	12.5	7.8	6.2	4.7
3–4 days/week	5.0	5.1	11.7	5.6	8.0	5.1
5–6 days/week	15.7	12.1	15.7	12.0	19.1	11.5
Everyday	69.8	76.7	60.0	74.7	66.8	78.7
Snacks						
Less than 1 time/week	8.7	8.1	6.0	5.6	6.4	4.9
1–3 times/week	11.9	12.9	13.1	14.2	13.5	12.9
4–6 times/week	65.2	67.6	65.8	67.4	70.6	70.2
1 time/day and above	14.1	11.5	15.1	12.8	9.5	12.0
Beverages ^#,@@^						
Less than 1 time/week	6.6	6.7	3.2	4.3	2.0	2.2
1–3 times/week	8.5	9.9	9.5	9.7	6.6	5.9
4–6 times/week	65.0	68.3	65.0	67.4	69.0	69.0
1 time/day and above	19.9	15.1	22.3	18.6	22.5	22.9
Plain water ^#,@^						
Less than 1 cup/day	5.2	1.6	2.8	1.9	2.0	2.4
1–2 cups/day	25.6	22.3	18.1	19.3	19.9	14.1
3–4 cups/day	27.0	31.0	28.8	25.0	37.0	31.2
5 cups/day and above	42.1	45.0	50.3	53.9	41.2	52.4

Note: Multilevel model was used to evaluate the effect of the intervention after adjusting the nutrition knowledge score. Group effect: ^#^ *p* < 0.05; time effect: ^@^ one year *p* < 0.05; ^@@^ two years *p* < 0.05; time × group effects: * *p* < 0.05.

**Table 8 nutrients-14-03997-t008:** Test of the fixed effects of various factors of dietary behavior score.

Effect	Type	*β* *	SE ^#^	T-Value	*p*-Value
Intercept			0.076	108.69	<0.001
Group	Intervention		0.088	2.15	0.032
	Control	Ref.			
Time	Second Year		0.100	2.67	0.008
	First Year		0.102	1.01	0.311
	Baseline	Ref.			
Time × Group	Second Year × Intervention		0.114	1.67	0.094
	Second Year × Control				
	First Year × Intervention		0.116	0.66	0.510
	First Year × Control				
	Baseline × Intervention	Ref.			
	Baseline × Control				

Note: After adjusting for the nutrition knowledge score, group effect: *p* > 0.05; time effect: *p* > 0.05; time × group effect: *p* > 0.05. * *β*, coefficient. ^#^ SE, standard error.

**Table 9 nutrients-14-03997-t009:** Comparison of height, weight, and BMI of children between intervention group and control group.

	Baseline		First Year		Second Year		Change (Baseline-Second Year)	
	Control	Intervention	Control	Intervention	Control	Intervention	Control	Intervention
Height (cm) ^@,@@^								
Male	127.42 ± 6.83	128.10 ± 6.93	133.37 ± 7.97	133.19 ± 9.09	141.06 ± 9.27	141.26 ± 8.86	6.79 ± 8.07	6.58 ± 7.07
Female	127.70 ± 8.10	127.26 ± 7.73	134.34 ± 9.31	133.77 ± 8.64	142.61 ± 10.03	142.52 ± 8.82	7.38 ± 8.02	7.65 ± 8.10
Weight (kg) ^#,@,@@^								
Male	25.99 ± 5.17	27.29 ± 6.58	29.32 ± 7.02	30.38 ± 12.35	35.59 ± 8.89	36.74 ± 9.46	4.79 ± 6.17	4.71 ± 5.55
Female	25.32 ± 5.47	25.82 ± 5.90	28.85 ± 6.86	29.07 ± 7.28	36.11 ± 8.65	36.57 ± 8.94	5.35 ± 6.27	5.40 ± 6.20
BMI (kg/m^2^) ^@@^								
Male	15.91 ± 2.12	16.48 ± 2.85	16.32 ± 2.54	18.31 ± 2.16	17.68 ± 2.86	18.18 ± 3.14	0.88 ± 1.54	0.85 ± 1.38
Female	15.38 ± 1.99	15.78 ± 2.37	15.79 ± 2.18	16.05 ± 2.66	17.53 ± 2.60	17.77 ± 2.91	1.07 ± 1.54	1.00 ± 1.44

Note: Multilevel model was used to evaluate the effect of the intervention after adjusting for physical activity. Group effect: ^#^ *p* < 0.05; time effect: ^@^ one year *p* < 0.05; ^@@^ two years *p* < 0.05. Note: Values are presented as mean ± SD.

## Data Availability

Data available on request due to privacy restrictions. The data presented in this study are available on request from the corresponding author. The data are not publicly available due to privacy.

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
