# Peer review of "Effect of School-Based Nutrition and Health Education for Rural Chinese Children"

_nutrients, 2022, doi:10.3390/nu14193997_

Round 1
Reviewer 1 Report
The article entitled "Effect of school-based nutrition and health education for rural 2 Chinese children" deals with the effect of a pedagogical protocol on learning good nutrition in Chinese schools.
The article is social science focused, rather than the area of biomedical science.
This can be seen in the surveys, which are very interesting but only show social data focused on learning. The intervention is relevant and shows significant data. However, they only show data in table 9 in relation to physiological changes, for example in growth. These changes may be due to genetic heterogeneity or malnutrition that have not been proven scientifically. They should have done tests to confirm if there is malnutrition and check the phenotype of the individuals to determine the development of each individual of the study. Likewise, they could have studied the incidence of disease in the individuals. However, they have won knowledge does not imply that they have applied it, in the end that will depend on the parents to a great extent.
The number of study subjects recruited is perfect. The methodology for the learning intervention is not described, this should be added detailed in supplementary material. There are questions like "obese children are more prone to hypertension, hyperlipidemia and other diseases" that are too complex for children of these ages to understand.
I encourage the authors to publish in more social journals, since I imagine it will be difficult to harvest samples to perform biomolecular tests.
Author Response
Dear reviewer,
Thank you for your letter and comments concerning our manuscript entitled "Effect of school-based nutrition and health education for rural Chinese children"(nutrients-1913792). Based on your suggestions, we have accordingly revised our manuscript. Below you could find the point-to-point response to the questions regarding the manuscript.
We hope that our answers have satisfied your comments and look forward to your response.
Response to Reviewer 1 Comments
Point 1: The article is social science focused, rather than the area of biomedical science.
Response 1: Thanks to Reviewer for reminder. We collected indicators related to nutrition education such as nutrition knowledge, eating behaviors and indicators related to nutrition status such as height and weight of children. This study mainly analyzes the effect of nutrition and health education, which belongs to the area of nutritional education for health promotion. There are many articles of the same type published in nutrients, as follows:
- El Harake MD, Kharroubi S, Hamadeh SK, Jomaa L. Impact of a Pilot School-Based Nutrition Intervention on Dietary Knowledge, Attitudes, Behavior and Nutritional Status of Syrian Refugee Children in the Bekaa, Lebanon. Nutrients. 2018;10(7):913. Published 2018 Jul 17. doi:10.3390/nu10070913
- Hamulka J, Wadolowska L, Hoffmann M, Kowalkowska J, Gutkowska K. Effect of an Education Program on Nutrition Knowledge, Attitudes toward Nutrition, Diet Quality, Lifestyle, and Body Composition in Polish Teenagers. The ABC of Healthy Eating Project: Design, Protocol, and Methodology. Nutrients. 2018;10(10):1439. Published 2018 Oct 5. doi:10.3390/nu10101439
- Kaufman-Shriqui V, Fraser D, Friger M, et al. Effect of a School-Based Intervention on Nutritional Knowledge and Habits of Low-Socioeconomic School Children in Israel: A Cluster-Randomized Controlled Trial. Nutrients. 2016;8(4):234. Published 2016 Apr 21. doi:10.3390/nu8040234
Point 2: However, they only show data in table 9 in relation to physiological changes, for example in growth. These changes may be due to genetic heterogeneity or malnutrition that have not been proven scientifically. They should have done tests to confirm if there is malnutrition and check the phenotype of the individuals to determine the development of each individual of the study.
Response 2: Thanks for your suggestion. We assessed malnutrition of children at baseline, the 1st and 2nd years according to the Chinese Screening Standard for Malnutrition in School Children and Adolescents (WS/T456-2014). Malnutrition rates declined in both the intervention and control groups. The different results may be related to the lack of intervention for other factors and confounders in our study, and the reasons for these changes need further study.
Point 3: However, they have won knowledge does not imply that they have applied it, in the end that will depend on the parents to a great extent.
Response 3: Thanks for your advice. Whether children apply what they learned may be influenced by many factors such as family and societal factors, but the level of knowledge is one of the most important factors. The dietary behaviors of children are affected by the level of nutrition knowledge. Our study found that nutritional knowledge related to breakfast and the frequency of eating breakfast improved after the intervention. However, the behavior of consuming snacks and beverages did not significantly improve after the intervention, which may require the joint efforts of schools, parents and society. Therefore, the influence of parents on children's knowledge application should be further studied. We have supplemented relevant contents in the discussion.
Point 4: The methodology for the learning intervention is not described, this should be added detailed in supplementary material.
Response 4: Thanks for your suggestion. The methodology for the learning intervention is described in 2.2 Intervention Methods. Nutrition education was mainly conducted for students by offering nutrition and health courses, which were part of students' daily courses. We also conducted unified training for teachers and used unified teaching materials for the courses. We have added relevant contents in 2.2 Intervention Methods.
Point 5: There are questions like "obese children are more prone to hypertension, hyperlipidemia and other diseases" that are too complex for children of these ages to understand.
Response 5: Thanks for your advice. We would reasonably set the difficulty of the questionnaire in future research.
Warm regards,
Reviewer 2 Report
Dear authors,
I enjoyed reading your manuscript. The topic is very important also in Europe.
The number of participants is adequate, however it is not clear if they were also able to reject their participation in the study. How many of them refused to participate? What were the reasons for dropping out?Could you add that information in the section 2.1?
Lines 135-138: Why are the sentences written in italic?
What is the volume of the bag of milk? Could you add that information in the manuscript so that it is clear to us who do not live in China? Could you add the metrics in the table 1? What is the size (grams, ml, etc) of the particular item in the table 1?
Otherwise, tables and figures are presented clearly.
In Discussion, line 246: change the order as you presented in methods section: First you checked nutrition knowledge, then dietary intake and lastly dietary behaviors.
Author Response
Dear reviewer,
Thank you for your letter and comments concerning our manuscript entitled "Effect of school-based nutrition and health education for rural Chinese children"(nutrients-1913792). Concern for the health of Chinese children is at the core of our work. It’s very glad to have your recognition of our work. We believe it is meaningful to improve the nutritional health of rural children. In the future studies, we will maintain our focus on the nutrition knowledge, dietary behaviors and nutritional status of rural children, and strive to contribute to the healthy growth and development of rural children.
Warm regards,
Response to Reviewer 2 Comments
Point 1: The number of participants is adequate, however it is not clear if they were also able to reject their participation in the study. How many of them refused to participate? What were the reasons for dropping out?Could you add that information in the section 2.1?
Response 1: Thanks for your advice. The participants had the right to refuse to participate in the study, but no one refused to participate. We integrated nutrition education courses into children's daily courses. Children were willing to learn new knowledge, so no child refused to participate in the study. We would add this information in the section 2.1.
Point 2: Lines 135-138: Why are the sentences written in italic?
Response 2: Thanks for your reminder. We would correct it.
Point 3: What is the volume of the bag of milk? Could you add that information in the manuscript so that it is clear to us who do not live in China? Could you add the metrics in the table 1? What is the size (grams, ml, etc) of the particular item in the table 1?
Response 3: Thanks for your advice. The volume of milk is 300ml. The weight of egg is 50 ~ 60g. The volume of a cup of water is about 250ml. We would add this information in table1.
Point 4: In Discussion, line 246: change the order as you presented in methods section: First you checked nutrition knowledge, then dietary intake and lastly dietary behaviors.
Response 4: Thanks to Reviewer for reminder, we would correct it.
Round 2
